# Design of Folate-Containing Liposomal Nucleic Acid Delivery Systems for Antitumor Therapy

**DOI:** 10.3390/pharmaceutics15051400

**Published:** 2023-05-03

**Authors:** Elena V. Shmendel, Pavel A. Puchkov, Michael A. Maslov

**Affiliations:** Lomonosov Institute of Fine Chemical Technologies, MIREA—Russian Technological University, Vernadsky Ave. 86, 119571 Moscow, Russia

**Keywords:** folate, folate receptor, tumor-targeting, lipoconjugates, PEG, liposomes, gene delivery

## Abstract

The delivery of therapeutic nucleic acids is a prospective method for the treatment of both inherited and acquired diseases including cancer. To achieve maximal delivery efficiency and selectivity, nucleic acids should be targeted to the cells of interest. In the case of cancer, such targeting may be provided through folate receptors overexpressed in many tumor cells. For this purpose, folic acid and its lipoconjugates are used. Compared to other targeting ligands, folic acid provides low immunogenicity, rapid tumor penetration, high affinity to a wide range of tumors, chemical stability, and easy production. Different delivery systems can utilize targeting by folate ligand including liposomal forms of anticancer drugs, viruses, and lipid and polymer nanoparticles. This review focuses on the liposomal gene delivery systems that provide targeted nucleic acid transport into tumor cells due to folate lipoconjugates. Moreover, important development step, such as rational design of lipoconjugates, folic acid content, size, and ζ-potential of lipoplexes are discussed.

## 1. Introduction

Gene malfunction causes either overexpression or downregulated expression of proteins resulting in many inherited and acquired diseases including cancer. Nucleic acids (NA) of different types are used as therapeutic agents. The delivery of plasmid DNA (pDNA) or messenger RNA (mRNA) leads to the expression of the missing protein in target cells. Overexpressed proteins can be silenced by means of double-stranded small interfering RNA (siRNA). The antisense strand of siRNA is a pharmacologically active component, while the sense strand facilitates transport of siRNA to the intracellular RNA-endonuclease Ago2 [1].

Upon intravenous administration, NA trafficking encounters such barriers as (1) interaction with blood components and destruction by endonucleases; (2) rapid renal clearance (especially for short NA molecules); (3) lack of selective accumulation in target cells; (4) inefficient cellular penetration due to electrostatic repulsion between phosphate NA groups and negatively charged cell membrane; (5) inefficient endosomal escape to the cytoplasm; (6) nuclear penetration in case of pDNA delivery. Overcoming the aforementioned barriers requires a delivery system capable of targeted NA transport into tumor cells [2].

Unlike normal tissues, rapidly growing or undifferentiated tumors overexpressed folate receptors (FR) [3]. For example, upregulated FR expression found in squamous-cell carcinomas (KB cells contain 4 million FR per cell) [4,5], lung and pancreatic adenocarcinomas [6], ovarian (IGROV-1, SKOV-3) [5,7], cervical (HeLa) [5,8], colorectal [9], ependymal brain [10], kidney (including HEK293) cancer [11]. FR are overexpressed on cell membranes of non-small lung cancer, stromal tumor-associated macrophages [12], and malignant melanoma [13,14]. Tumor tissue from seventy-one percent of patients with triple negative breast cancer showed FR overexpression [15]. FR overexpression is also observed for choriocarcinoma [5], meningioma [16], osteosarcoma [17], and non-Hodgkin lymphoma [8]. Four FR isoforms are known: FRα, FRβ, FRγ, FRδ. The alpha isoform is the most abundant and the most studied FR, which not only mediates folate internalization, but also participates in cancer signaling regulation [18]. Moreover, FRα has a minimal physiological role in normal tissues making it a potent anti-tumor target [19].

Targeted NA delivery can be achieved with delivery systems containing folic acid (FA, Figure 1) or its derivatives due to its high affinity for overexpressed FR (primarily the alpha isoform) on the surface of tumor cells. FA is a water-soluble vitamin B_9_ required in nucleotide synthesis during proliferation of any cell. FA also participates in DNA methylation as well as synthesis of S-adenosylmethionin, histones, G-proteins, and many metabolic building blocks [20]. FA internalized by cells through the reduced folate carrier (RFC), the proton-coupled folate transporter (PCFT) or FR. Only the last one is capable of binding FA conjugates and transport them by receptor-mediated endocytosis [18,21].

FA consists of three distinct structural units: 2-amino-6-methylpteridin-4-one (I), *p*-aminobenzoic (II) and L-glutamic (III) acids (Figure 1).

The advantages of FA-mediated cancer cell targeting compared with other targeting ligands such antibodies or (poly)peptides include: (1) low immunogenicity allowing repeated administration; (2) rapid tumor penetration due to low molecular weight of FA conjugates; (3) high affinity to a wide range of tumors; (4) high stability allowing use of organic solvents, heating and prolonged storage time; (5) simple chemical structure providing cheap production and ease of quality control [22,23,24].

Different delivery systems can utilize targeting by FA ligand. For example, targeted small molecule-drug conjugates [19,21] as well as liposomal formulations of drugs (doxorubicin, daunorubicin, vintafolide, etc.) containing FA lipoconjugates [24,25,26,27] demonstrated enhanced antitumor activity, while FA conjugated polymers have also been used for cancer targeting [22,28,29,30]. Such drug delivery systems are generally of low-cost and are less toxic antitumor therapies than traditional chemotherapeutic drugs. Another delivery approach entails the conjugation of folic acid to engineered viruses, which results in significant and selective binding to target cells. But transduction efficiency of folate-containing viruses is low, probably due to incompatible viral tropism to target cells [31]. Oncolytic viruses are capable of proliferating in tumor cells and of their destruction [32]. Nevertheless, disadvantages associated with this approach include: immunogenicity, mutagenicity and toxicity [33]. FA-conjugated inorganic nanoparticles (silica, metal, graphene, etc.) have also been used to deliver nucleic acid cargos [28,34,35].This review focuses on the delivery of FA conjugated NA as well as folate-containing gene delivery systems that provide targeted NA transport into tumor cells.

## 2. Conjugation of Folic Acid to Nucleic Acids

FA covalently binding to NA presents a self-aggregating system targeted for tumor cells. Click reactions are often used for conjugation, for example alkyne-azide reactions via [3+2]-cycloaddition. Early works demonstrated FA attached via triethylene glycol spacer to the 3′-end of the sense strand of siRNA (Figure 2A) provided 40–60% gene silencing in hard-to-transfect RBL-2H3 cells [36]. FA binding to the 5′-end of the sense strand of siRNA via a peptide spacer (Figure 2B) resulted in targeted delivery of FA-siRNA conjugates to KB tumor xenografts in mice in vivo, but gene silencing activity was low [37]. Poor transfection activity of FA-siRNA conjugates was overcome using a polymer delivery system which provided more than 50% gene knockdown in vitro [38]. The conjugation of FA to the center of the sense strand of anti-luciferase siRNA (Figure 2C) increased silencing of the exogenous gene in HeLa cells up to 80%. Moreover, treatment of HeLa cells expressing *Bcl-2* protein with FA-siRNA conjugates provided 72% knockdown of the endogenous anti-apoptotic gene Bcl-2 [39].

Despite promising in vitro results, in vivo efficiency of FA-siRNA conjugates has not been demonstrated due to lack of endosomal escape functionality [37]. Secondly, any chemical modification may decrease the biological activity of the therapeutic NA. Thirdly, upon in vivo administration, NA is inactivated by serum nucleases. Hence, an efficient delivery system is required to target NA transport into tumor cells and to exhibit therapeutic activity in vivo.

## 3. Noncovalent Interaction between Folic Acid and Nonviral Delivery Systems

To increase transfection efficiency FA may be incorporated into NA delivery systems through noncovalent interactions in a few ways: (1) by adding free FA into culture media [40]; (2) by mixing of FA and NA with subsequent addition of nonviral delivery system [40]; (3) by adding free FA to preformed complexes of NA and nonviral delivery systems [41,42,43,44] (Table 1). Noncovalent incorporation of FA into nonviral delivery systems was studied in cell lines with upregulated FR expression (KB, human epithelial carcinoma; HeLa; NCI-H460, human large-cell lung carcinoma) as well as in cells with normal FR expression (F98, human glioma cell line; CHO, chinese hamster ovary; A549, lung carcinoma epithelial cells).
pharmaceutics-15-01400-t001_Table 1Table 1Nonviral NA delivery systems with noncovalent binding FA.Delivered NADelivery SystemFA IncorporationOptimized FA AmountCell LineRef.pCMV-luc pDNAPEI ^1^adding to culture media1 mMKB, F98, CHO[40]mixing of FA and NA with subsequent complexation0.5 mMadding to preformed complexesCharge ratio FA/PEI/pDNA 60:8:1B16-F10[41]survivin siRNAPEI-SS ^1^adding to preformed complexesCharge ratio FA/PEI-SS/siRNA 20:8:1HeLa, A549[42]pGFP-N2 and pGL3 pDNADDCTMA ^1^–CholOH ^1^ (3:1)adding to preformed complexes100 μg/mLNCI-H460[43]pEGFP-C2 pDNA2X3 ^1^–DOPE ^1^ (1:2)adding to preformed complexes1 mMKB-3-1[44]^1^ compound structures presented in Figure 3.
Figure 3Polymers, dextrans, and lipids as nonviral NA delivery systems.
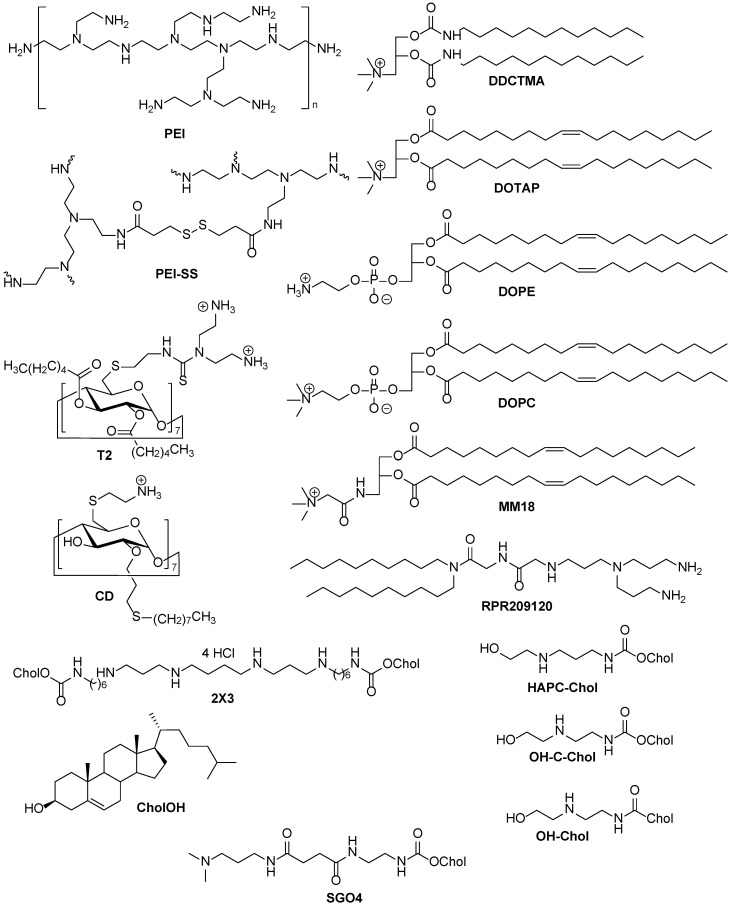


The introduction of free FA (1 mM) into culture media resulted in a 124-fold increase in the efficiency of transfection mediated by PEI-pDNA complexes (Figure 3) in KB cells [40] (Table 1). The same results were obtained on cells with normal FR expression (F98 and CHO cells) due to FA inhibition of serum, rather than receptor-mediated endocytosis, in the case of PEI delivery system.

In the case of preliminary mixing of FA and pDNA with subsequent addition of PEI optimal FA concentration was lower (0.5 mM) [40] (Table 1). Unlike other anionic molecules such as cholic acid, glutamic acid, citric acid, and EDTA, only FA increased transfection efficiency of PEI-pDNA complexes.

A similar system was prepared by adding of free FA to preformed complexes [41] (Table 1). Complexes with low amount of FA (charge ratio 15:8:1 and 30:8:1) showed significantly lower transfection efficiency than others probably due to rapid particle aggregation (ζ-potential was near 0 mV). Higher amounts of FA lead to more stable particles with charge ratio 60:8:1 having the most optimal size (~260 nm), cytotoxicity and transfection efficiency together. Complexes with charge ratio 60:8:1 were used for in vivo delivery of pCMV-luc into ddY mice and showed significantly higher gene expression in the liver, kidney, spleen, and lung compared to PEI/pDNA complexes without FA. Preincubation of mices with free FA led to a decreased transfection efficiency indicating receptor-mediated uptake.

Stimuli-responsive complexes of PEI-SS (Figure 3) and noncovalently bound FA delivered siRNA effectively, but no differences between HeLa cells with upregulated FR expression and A549 with normal FR expression were observed [42].

In the case of pDNA delivery, lipoplexes based on liposomes DDCTMA–CholOH [43] or 2X3–DOPE [44] (Table 1, Figure 3) with noncovalently bound FA rather than without FA were more effective. The experiments were performed on cell lines with upregulated FR expression (NCI-H460, KB-3-1). Preincubation of these cells with free FA decreased transfection efficiency at a low N/P ratio (2:1, number of cationic amino groups of lipids per phosphate groups of NA) but did not affect the NA delivery at the N/P ratio of 4:1. Therefore, excessive positive charge at high N/P ratios allows lipoplexes to be delivered by nonspecific endocytosis, while FRs are blocked by free FA [44].

Enhanced transfection efficiency was also observed with noncovalent FA–cyclodextrin T2 (Figure 3) complexes compared with their non-targeted counterparts. Notably, the same results were achieved in vivo: noncovalent FA–cyclodextrin T2 complexes provided enhanced gene expression in the liver and lungs of mice [45].

It should be noted that nonviral delivery systems with noncovalent binding FA are simpler and cheaper compositions for NA delivery compared with targeted systems containing FA lipoconjugates. However, the most commonly described FR-targeted delivery systems are liposomes decorated with covalent lipoconjugates of FA.

## 4. Folate Lipoconjugates as Components for Liposomal Delivery System

Cationic liposomes as nonviral delivery systems protect NA from interaction with biological media components, preventing NA degradation and increasing its accumulation in target organs and cells [46,47]. To achieve targeted NA delivery, liposomes should bind with receptors on the surface of target cells.

FA lipoconjugates may be incorporated in two ways: (1) adding of FA lipoconjugate to preformed NA-liposome complexes (lipoplexes) [48]; (2) inclusion of FA lipoconjugate in the liposomal formulation with subsequent lipoplex formation [11,44,49].

### 4.1. FA Lipoconjugate Structure

FA lipoconjugates used in liposomal delivery systems usually feature FA residues covalently bound to a hydrophobic domain through different spacers and linkers (Figure 4). It should be noted that FR correctly recognizes the FA residue only if the latter is connected to the spacer by γ-carboxyl group [50]. 1,2-Distearoyl-*sn*-glycero-3-phosphatidylethanolamine (DSPE) is the most popular compound used as a hydrophobic domain anchoring FA lipoconjugate in the liposomal bilayer [48,51,52,53]. Besides DSPE, cholesterol (CholOH) [54,55], ditetradecylglycerol (Dg) [11,44,49], and diethers [23] may be used as hydrophobic domains.

A spacer (Figure 4) ensures the availability of the FA moiety for FR recognition and binding. The most common spacer is a polyethylene glycol (PEG) chain which could protect the lipoplexes from nonspecific interactions with the components of blood serum, mask their positive charge, and prevent self-aggregation [56,57,58] and thus facilitate better transfection efficiency [59]. However, PEG chains could prevent endosomal escape of the therapeutic load [60], thereby drastically decrease in vitro transfection [61], and sometimes cause accelerated blood clearance [62,63,64,65]. A molecular weight of PEG chains may varied from 570 to 5000 Da [23,51,66,67,68]. Nonetheless, PEG spacer with a high molecular weight of 5000 Da inhibited cellular uptake of siRNA lipoplexes compared to a short analog (2000 Da) [67]. In this case, lower amounts of the FA lipoconjugate are required [69]. Besides PEG, other hydrophobic (hexamethyl or longer alkyl chains) or hydrophilic (butoxybutoxybutyl) spacers may be used [44,49], providing effective NA delivery in vitro.

A linker (Figure 4) provides binding between all structure elements of FA lipoconjugates and affects the physicochemical properties of liposomes into which they have been formulated. For example, FA may be connected with the spacer by a cysteine residue [51,70] or a diaminopropionyl-aspartyl-cysteine with maleimide group [68], providing better solubility of the FA lipoconjugate and better mobility of the FA residue. A biodegradable peptide linker between FA and a spacer as well as a carbamoyl linker between a spacer and a hydrophobic domain are the most common ones [11,44,49,53]. For a more convenient synthesis, a linker based on maleimide may be used for the binding of FA and a spacer [48].

### 4.2. Optimal Content of FA Lipoconjugates with Different PEG Spacers in Cationic Liposomes

Despite the contradictory properties of PEG, it is the most common spacer used in FA lipoconjugates. The PEG spacer between FA and the lipid part of the conjugate is assumed to prevent coating of FA by serum proteins and therefore to allow proper binding with FR [71].

The transfection efficiency depends on the content of FA lipoconjugate within the liposomal formulation. Early work demonstrated that the optimal content of FA lipoconjugate with PEG_3400_-spacer ranged from 0.03% to 0.1% mol. during experiments on FR-positive M109 cells (Table 2), and was 0.01% in the case of in vivo experiments on L1210A intraperitoneal tumor model [68]. Also, FA lipoconjugate content higher than 1% might cause multimerization preventing FA-FR binding [72]. FA lipoconjugate based on cholesterol (Chol-suc-PEG_2000_-FA) provided the best transfection efficiency at 0.1% mol. from the range 0.01–1.0% in vitro [55].

The optimal content of FA lipoconjugate within liposomal formulation may be determined by the structure of cationic lipid as well as the length of PEG spacer. For example, optimal content of DSPE-PEG_2000_-FA lipoconjugate was 1% mol. in the case of liposomes OH-Chol–DOPE (3:1 mol.) containing a lipid with an amide linker (Table 2). After the replacement of the amide linker with a carbamoyl one (lipids OH-C-Chol and HAPC-Chol) optimal content of DSPE-PEG_2000_-FA changed to 2% mol., whereas suppression of luciferase activity significantly increased. It should be noted that all three cationic liposomes with 3% mol. content of DSPE-PEG_2000_-FA demonstrated the worst transfection of KB-Luc cells [67]. But no correlation between in vitro and in vivo experiments was found. Moreover, using of 2 mol% DSPE-PEG_5000_-FA lipoconjugate with liposomes HAPC-Chol–DOPE (3:1 mol.) resulted in decreased siRNA delivery efficiency, indicating that a longer PEG chain (5000 Da instead of 2000 Da) could inhibit FR-mediated uptake by cells. A possible explanation is the greater probability of PEG_5000_ chains to interact between each other which lead to “masking” of the ligand [72]. On the other hand, cationic liposomes containing 0.5 mol% DSPE-PEG_5000_-FA provided more effective siRNA delivery both in vitro and in vivo compared with liposomes containing FA lipoconjugate with shorter PEG spacers [69]. Moreover, liposomes containing 0.5 mol% Chol-suc-PEG_4000_-FA demonstrated a three-fold higher cellular uptake by HepG2 cells compared with liposomes containing Chol-suc-PEG_2000_-FA with shorter PEG spacer [73].

In the literature, cationic liposomes containing 0.01% mol. [66], 1% mol. [74], 1.5% mol. [75] of DSPE-PEG_2000_-FA are described, but no comparison has been carried out between different contents of the FA lipoconjugate.

FITC-ODN delivery by targeted cationic liposomes 2X3–DOPE (2:1 mol.) showed the optimal content of Dg-PEG_800_-FA lipoconjugate of 2% mol. (Table 2) [49]. At low N/P ratio, FA-targeted liposomes were more effective than both conventional liposomes and Lipofectamine 2000 in terms of their capacity to deliver FITC-ODN and pDNA into KB-3-1 and HEK 293 cells [44,49]. The decrease in transfection efficiency of FA-containing liposome in the presence of free FA suggests a receptor-mediated delivery of pDNA by this formulation [11]. Also, the same liposomal composition effectively delivered antisense oligonucleotides, siRNA, including in in vivo experiments [11,76,77,78].

Cationic liposomes based on MM18 and diether-PEG_570_-FA lipoconjugate delivered pDNA more effectively at 9:1 wt [23].

Analyzing the aforementioned data, we suggest that the shorter PEG spacer the higher the FA lipoconjugate content required for effective targeting and NA delivery in vitro. Short spacers seem to have limited mobility, which requires compensation in the form of increased FA “density” to be recognized by FR.

### 4.3. Optimal Physicochemical Properties of Folate-Containing Liposomal NA Delivery Systems

Lipoplexes are complexes between nucleic acids and cationic lipids. The formation of the lipoplexes occurs as a result of electrostatic interactions between the polar head groups of cationic lipids and phosphate groups of NA. The stability of lipoplexes depends on the ionic strength of the medium in which they are formed, and some lipoplexes may prematurely release encapsulated NA under physiological conditions.

The structure of lipoplexes can be defined on two lengths. In the range of 10–50 nm, the structure is set by thermodynamic equilibrium and are therefore highly regular, whereas larger-scale characteristics are set by kinetics and are therefore highly polydisperse. Optimizing the transfection efficiency of lipoplexes requires the ability to control their properties, since size and surface charge determine their stability in serum, their circulation time in vivo, and their interactions with cells [79].

The fine structure of lipoplexes depends on both the type of lipids and the type of NA. In the case of the double helix DNA most common complexes are multilamellar phase and normal and inverted hexagonal phase structures [80]. Shorter siRNA could form lamellar and bicontinuous cubic phases, while the inverted hexagonal complexes had high cytotoxicity. Cationic gemini surfactants interact with siRNA and could form condensed micellar phase with siRNA sandwiched between surfactant micelles.

The presence of PEG-lipoconjugates in the lipoplexes forms sterical shield, making the complexes more protected when interacting with serum components, but can block important interactions of complexes with cellular membranes. Cationic liposomes incorporating PEG_2000_-lipoconjugates form with DNA lipoplexes with lamellar structure [81].

To achieve targeted NA delivery by cationic liposomes, size, ζ-potential and N/P ratio must be optimized. To date, no clear correlation has been found between the size of lipoplexes and their transfection efficiency, however, the optimal size is known to lie in the 10–200 nm size range, beyond which nanoparticles are rapidly eliminated from the organism [82]. All other factors being equal, one should choose lipoplexes with smaller particle diameters [67].

ζ-Potential is another important parameter in the NA delivery. A positive charge allows cationic liposomes to bind NA into stable lipoplexes. Positively-charged lipoplexes may be effectively internalized by cells due to electrostatic interactions with the negatively-charged plasma membrane. However, such nonspecific uptake interferes with selective NA delivery to target cells. Hence, targeted lipoplexes should have ζ-potential approximately equal to zero to exclude both nonspecific endocytosis [11,66]. ζ-Potential of lipoplexes may be tuned by changing the N/P ratio or by coating with FA and/or PEG [25,62].

Lipoplexes containing cholesterol-based cationic lipid SGO4, helper lipid DOPE, DSPE-PEG_2000_, and its FA-conjugate were prepared and used for in vitro delivery of pDNA encoding the luciferase gene (Table 3). The transfection efficiency was not affected by the N/P ratio (from 2:1 to 4:1), although NA binding was incomplete at the N/P ratio of 2:1. Folate-containing lipoplexes transfected FR-positive cells (HeLa and KB) more effectively compared to control lipoplexes without FA lipoconjugate. Moreover, preincubation of cells with free FA decreased transfection efficiency, indicating specific lipoplex uptake through FR [66].

The highest transfection efficiency of lipoplexes MM18–Diether-FA-PEG_570_ containing 10% of FA lipoconjugate with a short PEG spacer was observed at N/P ratios of 1:1 and 2:1 in vitro [23]. Notably, these two compositions differed drastically in their physicochemical parameters: lipoplexes at the N/P ratio of 1:1 were much larger (521 nm) and displayed a strongly negative ζ-potential (−49 mV) compared with lipoplexes at the N/P ratio of 2:1 that were considerably smaller (164 nm) and exhibited a strongly positive ζ-potential (+47 mV). The presence of high concentrations of free FA in cellular media (more than 10 nM) significantly decreased the transfection efficiency at the N/P ratio of 1:1, while lipoplexes with a higher N/P ratio of 2:1 were less affected, probably due to nonspecific endocytosis. Also, lipoplexes prepared at the N/P ratio of 1:1 were tested in vivo. After local nasal administration, one mouse exhibited a positive bioluminescent signal. Luciferase expression was observed in the trachea homogenates of the two mice (Table 3).

Lipoplexes with positive ζ-potential have also used for pDNA delivery [9,55,83]. During in vitro experiments, DOTAP based lipoplexes (Table 3) showed increased transfection efficiency on FR-positive cells (SKOV-3, KB, and HeLa), while no enhancement was observed on FR-negative HepG2 cells compared to control lipoplexes without FA-containing component. It should be noted that the percentage of transfected cells was low in the case of both SKOV-3 and KB cells (15 and 30%, respectively). In vivo delivery of both pPEDF (pigment epithelium-derived factor gene) into HeLa xenografts [83] and pIL15 (interleukin-15 gene) into CT26 xenografts [9] suppressed angiogenesis and tumor proliferation better than their non-targeting counterparts. Unfortunately, selectivity of pDNA delivery by these positively-charged lipoplexes was not studied.

The same liposomal formulations were used for targeted genome editing based on CRISPR-Cas9 technology. Folate-containing liposomes formed stable lipoplexes with the ovarian cancer-related DNA methyltransferase 1 (gDNMT1) plasmid at N/P ratios of 12:1 and 16:1. N/P ratios are much higher than that used previously for pEGFP delivery due to the large size of the gDNMT1 plasmid (8500 bp). FA-containing lipoplexes successfully inhibited DNMT1 expression in vitro and produce targeted genome editing based on CRISPR-Cas9 technology. FA-containing lipoplexes induced necrosis of cancer cells in vivo and exhibited favorable antitumor efficiency comparable with paclitaxel in a mouse model of ovarian cancer [84].

A slightly different formulation (DSPE-based PEG lipid instead of mPEG_2000_-Chol) was used for co-delivery of Bmi1 siRNA with ursolic acid. Lipoplexes demonstrated synergistic antineoplastic effect on KB cells in vitro and in vivo (xenografts) [85]. After the treatment 71% inhibition of KB cells was observed (Table 3). It should be noted, that folate-containing lipoplexes showed stronger gene inhibition than lipoplexes without FA-conjugate. Co-delivery of Bmi1 siRNA with ursolic acid by folate-containing liposomes provided significant tumor growth inhibition compared to different control groups.

A series of lipoplexes containing cholesterol-based cationic lipid (HAPC-Chol, OH-Chol or OH-C-Chol), DOPE, and the FA derivative of DSPE-PEG_2000_ was developed [67]. They had similar sizes around 180 nm, positive ζ-potentials and siRNA delivery efficiency on KB-EGFP cells (Table 3). It should be noted that transfection efficiency was also similar to non-targeting analogs, probably due to excessive positive charge provoking non-specific endocytosis. HAPC-Chol–DOPE–DSPE-PEG_2000_-FA composition with the smallest size was chosen for in vivo experiments on KB xenografts. But no differences were found in the suppression of tumor growth between FA-containing lipoplexes and the non-targeting PEGylated analog as was the case in in vitro studies and probably for the same reason: positive ζ-potential prevented targeting on FR [67].

FA-containing cationic liposomes were prepared and optimized for in vivo siRNA delivery [69]. Liposomes differed both in the length of PEG spacer (2000, 3400 or 5000 Da) and in the molar percentage of FA lipoconjugate. Maximal in vitro siRNA delivery was achieved by lipoplexes containing 0.5 mol% DSPE-PEG_5000_-FA, but this amount was not enough to prevent agglutination in vivo. Inclusion of 1–2 mol% FA-PEG-DSPE into siRNA lipoplexes might be needed for preventing agglutination during systemic injection. Therefore, 2.5 mol% PEG lipid was added to provide additional lipoplex stabilization in blood circulation, which permitted an increased accumulation of Cy5-labeled siRNA in KB xenografts [69].

Gladkikh et al. examined siRNA delivery efficiency of lipoplexes containing polycationic lipid 2X3, DOPE, and FA lipoconjugate on several tumor xenografts that exhibited different levels of FR expression [78]. The highest accumulation was observed in L1210 cells with maximal FR expression among cell lines used. MDR1-siRNA (against multidrug-resistant gene) delivery combined with polychemotherapy inhibited tumor growth, reduced necrosis and inflammation, and stimulated apoptosis in tumor cells without liver toxicity. Moreover, targeted delivery was achieved only by lipoplexes displaying ζ-potentials close to neutrality as previously described [61].

Cyclodextrin-based folate-targeted lipoplexes were used for in vitro delivery of siRNA targeted against RelA (a subunit of a central transcription factor NF-κB) overexpressed in prostate cancer cells [53]. This led to levels of knockdown of 44% for VCaP and 22% for LNCaP cells compared with non-targeted control. However, no differences between targeted and non-targeted formulations were observed for PC-3 cells, which have lower FR expression levels.

The use of an oligochitosan derivative of FA allowed the delivery of HIF-1α siRNA (against hypoxia-inducible factor-1α) to A375 cells with transfection efficiency comparable to Lipofectamine 2000. Also, transfection efficiency was two-fold higher compared with non-targeted control [86].

Recently, commercial transfectants (Lipofectamine 3000 and Fuse-It-DNA) were conjugated with FA via *N*-hydroxysuccinimide (NHS) chemistry for targeted pDNA delivery [87]. For this purpose, different amounts (0.4, 4, 12 mM) of NHS-PEG_5000_-FA were used. The lipoplexes showed a steady increase in particle size with increasing PEG_5000_-FA concentration, while ζ-potentials decreased in the opposite manner. The presence of 1 mM free FA in cellular media significantly decreased the transfection efficiency only for cancer cells overexpressing FR, verifying receptor-depended NA transfer. ca-Caspase3-pDNA delivery by FA-containing Lipofectamine 3000 led to a 1.5-fold increase in ca-Caspase3 activity in U87 cells compared with a non-targeted formulation. However, experiments conducted with FA-containing Fuse-It-DNA lipoplexes led to a ten-fold increase in ca-Caspase3 activity in MCF-7 cells.

## 5. Conclusions and Perspectives

Folic acid (FA) is an optimal targeting ligand providing selective NA delivery into tumor cells due to overexpressed FR. To develop an effective nonviral delivery system, the following points should be considered: (1) rational design of FA lipoconjugate; (2) optimization of lipoconjugate content; (3) optimization of transfection conditions including size and ζ-potential of lipoplexes as well as optimal N/P ratio.

The FA lipoconjugate should have a hydrophobic domain based on DSPE, cholesterol, or ditetradecylglycerol for anchoring in the liposomal bilayer. The most important part is the spacer binding FA to the rest of the molecule. The PEG spacer is optimal here, favoring a proper FA-FR interaction [71]. In general, the shorter the PEG spacer the higher the FA lipoconjugate content required for effective targeting and NA delivery.

In addition to the structure of the FA lipoconjugate, cationic lipid providing NA complexation is the other important component of cationic liposomes. The most common lipids used are monocationic [67] or polycationic derivatives of cholesterol [11,44,49] (Figure 3). Cationic lipids also determine association efficiency with target cells [67].

Nonviral delivery systems usually require a helper lipid. DOPE is often used for this purpose, providing effective endosomal escape of NA due to the inverted hexagonal phase formation in response to a change in endosomal pH value [88]. Thus, liposomes composed of cationic derivatives of cholesterol, helper lipid DOPE and FA lipoconjugate are optimal for targeted NA delivery into the tumor cells.

Finally, physicochemical parameters must be optimized. In this regard, lipoplex size should lie in the range 10–200 nm [82], while attention should be given to ζ-potential, as strong positive values may interfere with targeting due to non-specific endocytosis. This may be addressed by changing the N/P ratio of lipoplexes [11].

In summary, folate-containing nucleic acid delivery systems have great potential in targeted antitumor therapy providing optimized lipid structure and composition as well as physicochemical parameters. Moreover, further research may be concerning a combination of both therapy and diagnostics—theragnostics. In this case, fluorescently or radio labeled FA lipoconjugates should be developed [89,90,91]. As an alternative to chemically synthesized lipid vesicles, natural or engineered exosomes may be used [89]. They are easily decorated with FA to provide tumor-targeted NA delivery [92,93]. All delivery systems require further optimization to achieve applicable in vivo efficiency.

## Figures and Tables

**Figure 1 pharmaceutics-15-01400-f001:**
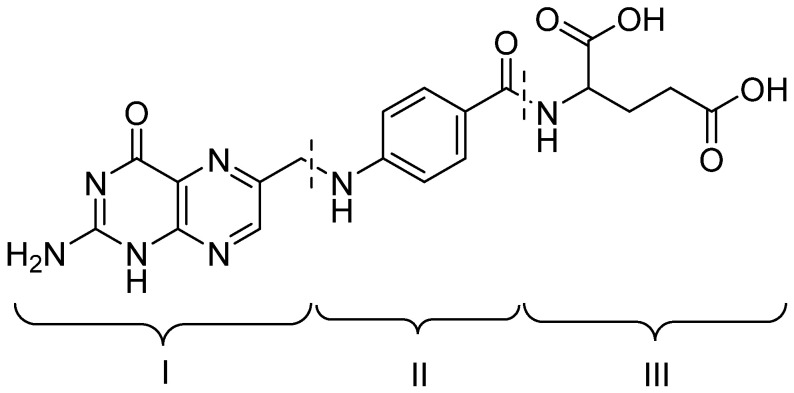
Structure of folic acid.

**Figure 2 pharmaceutics-15-01400-f002:**
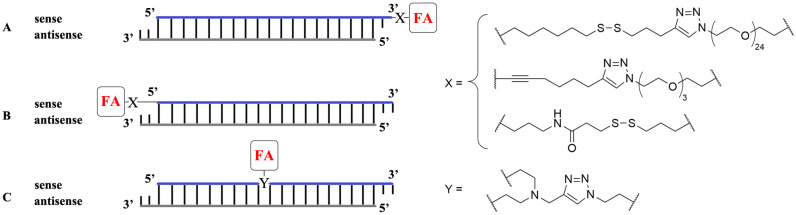
Conjugation of FA to the 3′-end of siRNA (**A**), to the 5′-end of siRNA (**B**) and to the center of siRNA (**C**).

**Figure 4 pharmaceutics-15-01400-f004:**
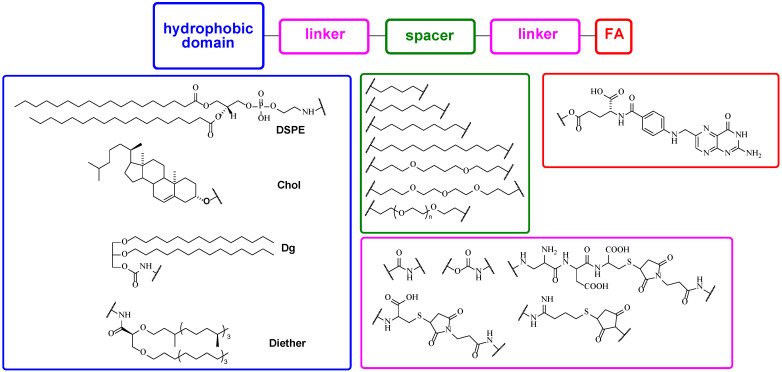
Typical structure elements of FA lipoconjugates.

**Table 2 pharmaceutics-15-01400-t002:** Examples of folate-containing cationic liposomes used.

Liposomal Composition	FA Lipoconjugate	Optimal Content of FA Lipoconjugate	NA	Target	Results	Ref.
DOPC ^1^–CholOH (65:35 mol.)	DSPE-PEG_3400_-Cys-FA	0.03% from the range 0.01–2.0%	pDNA contained the luciferase reporter gene under the CMV promoter.	FR-positive M109 cells	54 fluorescence units/mg cell lysate	[68]
RPR209120 ^1^–DOPE (1:1 mol.)		0.1% from the range 0.01–3.0%		FR-positive M109 cells	1500 ng luciferase/well	
		0.01% from the range 0.01–0.03%		L1210A intraperitoneal tumor model	125 ng/mg protein	
DOTAP ^1^–CholOH–mPEG_2000_-Chol (50:45:5)	Chol-suc-PEG_2000_-FA	0.1% from the range 0.01–1.0%	pEGFP	SKOV-3	14% cells	[55]
				KB	30% cells	
OH-Chol ^1^–DOPE (3:1 mol.)	DSPE-PEG_2000_-FA	1.0% from the range 1.0–3.0%	Luc siRNA/ EGFP siRNA	KB-Luc KB-EGFP	36% of untreated cells/ 75% EGFP expression level ^4^	[67]
OH-C-Chol ^1^–DOPE (3:1 mol.)		2.0% from the range 1.0–3.0%			18% of untreated cells/ 65% EGFP expression level ^4^	
HAPC-Chol ^1^–DOPE (3:1 mol.)		2.0% from the range 1.0–3.0%			20% of untreated cells/ 60% EGFP expression level ^4^	
2X3–DOPE (1:2 mol.)	Dg-PEG_800_-FA	2.0% from the range 0.5–2.0%	FITC-ODN	KB-3-1	86 RFU ^2^	[44,49]
				HEK293	82 RFU	
MM18 ^1^	Diether-PEG_570_-FA	10.0% from the range 2.0–10.0%	pDNA (pTG11033)	HeLa	1 × 10^7^ total RLU ^3^/mg of proteins	[23]

^1^ compound structures presented in Figure 3. ^2^ RFU: related fluorescence units. ^3^ RLU: related luminescence units. ^4^ The EGFP expression level (%) was calculated as relative to the fluorescence intensity of untransfected KB-EGFP cells.

**Table 3 pharmaceutics-15-01400-t003:** Parameters of folate-containing NA delivery systems used in vitro and in vivo.

Liposomal Composition	Lipoplexes				In Vitro		In Vivo	
	NA	N/P	Size (nm)	ζ-Potential (mV)	Results	Ref	Target	Ref.
SGO4–DOPE–DSPE-PEG_2000_–DSPE-PEG_2000_-FA (1:0.96:0.04:0.01 mol.)	pCMV-luc	3/1	147 ± 0.6	−2.09	1 × 10^7^ RLU ^4^/mg proteins (HeLa, KB) 1 × 10^6^ RLU ^4^/mg proteins (HEK293)	[66]	n.d. ^1^	
MM18–Diether-FA-PEG_570_ (9:1 wt)	pTG11033	1/1	521	−49	1 × 10^7^ RLU ^4^/mg proteins (HeLa, 16HBE14o) 1 × 10^6^ RLU ^4^/mg proteins (A549)	[23]	Swiss mice (nasal airway epithelium)	[23]
DOTAP–CholOH–mPEG_2000_-Chol–Chol-suc-PEG_2000_-FA (50:45:5:0.1 mol.)	pEGFP	5/1 ^2^	199.37 ± 2.56	37.97	30% of KB, 14% of SKOV-3	[55]	n.d. ^1^	
	pIL15	6/1 ^2^	250–300	18	n.d. ^1^		CT26 colon cancer mouse model	[9]
	pPEDF	6/1 ^2^	200	20	62% of HeLa	[83]	HeLa tumor xenografts	[83]
	gDNMT1 plasmid	12/1 ^2^	150	35	28.6% of indels (SKOV-3)	[84]	5-fold decrease of SKOV-3 tumor xenografts weight	[84]
DOTAP–CholOH–DSPE-mPEG_2000_– Chol-suc-PEG_2000_-FA (40:55:4.5:0.5 mol.)	Bmi1 siRNA	200/1 ^2^	165.1 ± 13.8	18.6	71% inhibition of KB cells	[85]	~4-fold decrease of KB tumor xenografts	[85]
OH-Chol–DOPE–DSPE-PEG_2000_-FA (3:1 mol.:1 mol%)	PLK1 siRNA	7/1	180.1 ± 2.8 ^3^	35.2	50% of untreatment (PKL mRNA level in KB-Luc cells)	[67]	n.d. ^1^	
OH-C-Chol–DOPE–DSPE-PEG_2000_-FA (3:1 mol.:2 mol%)			188.8 ± 1.2 ^3^	29.1	55% of untreatment (PKL mRNA level in KB-Luc cells)		n.d. ^1^	
HAPC-Chol–DOPE–DSPE-PEG_2000_-FA (3:1 mol.:2 mol%)			172.9 ± 1.2 ^3^	28.3	54% of untreatment (PKL mRNA level in KB-Luc cells)		KB tumor xenografts	[67]
DDAB–DOPE–PEG_1600_-Chol–DSPE-PEG_5000_-FA (1:1 mol.:2.5 mol%:0.5 mol%)		4/1	169.8 ± 1.6	39.3	10% of untreated KB-luc cells	[69]		[69]
2X3–DOPE–Dg-PEG_800_-FA (1:2 mol.:2 mol%)	MDR1 siRNA	1/1	175.2 ± 22.6	−3	n.d. ^1^		L1210, LLC, KB-3-1, KB-8-5 tumor xenografts	[78]
CD–DSPE-PEG_5000_-FA–DSPE-PEG_5000_-Methyl (2:1:1 mol.)	RelA siRNA	10/1	207.20 ± 5.39	4.78	44% reduction for VCaP cells, 22% reduction for LNCaP cells	[53]	n.d. ^1^	
Soybean lecithin S100–CholOH–oligochitosan derivative of FA (425 mg:75 mg:50 mg)	HIF-1α siRNA	-	95.3	2.41	550 mean fluorescence intensity on human MM cells (A375)	[86]	n.d. ^1^	

^1^ no data. ^2^ liposomes: pDNA mass ratio. ^3^ sizes determined for complexes with nonspecific siRNA. ^4^ RLU: related luminescence units.

## Data Availability

Not applicable.

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
