# Peer review of "Design of Folate-Containing Liposomal Nucleic Acid Delivery Systems for Antitumor Therapy"

_pharmaceutics, 2023, doi:10.3390/pharmaceutics15051400_

Round 1

Reviewer 1 Report

The title of the manuscript is perhaps inappropriate as the review has focused on liposome-mediated delivery. Lipid nanosystems should also embrace more recent developments in the field, such as the rise of solid lipid nanoparticles SLNs. There are several recently published works reporting folate decorated SLNs and their efficacy in nucleic acid (and drug) delivery. 

In general, the mini review could benefit from a more in depth treatment of the works cited. In its present form it lacks the authority to be considered a concise, yet critical and informative review of the subject matter given in the title. I have attached a file containing corrections and re-wordings and suggestions regarding the submitted manuscript to assist the authors. If it is indeed the intention of the authors to restrict the review to to liposome mediated delivery of drugs and NAs, then the title should be altered accordingly. 

Author Response

Dear Reviewers,

Thank you very much for profound analysis of our manuscript. According to your comments we carefully revised the article.

Reviewer 1

The title of the manuscript is perhaps inappropriate as the review has focused on liposome-mediated delivery. Lipid nanosystems should also embrace more recent developments in the field, such as the rise of solid lipid nanoparticles SLNs. There are several recently published works reporting folate decorated SLNs and their efficacy in nucleic acid (and drug) delivery. 

In general, the mini review could benefit from a more in depth treatment of the works cited. In its present form it lacks the authority to be considered a concise, yet critical and informative review of the subject matter given in the title. I have attached a file containing corrections and re-wordings and suggestions regarding the submitted manuscript to assist the authors. If it is indeed the intention of the authors to restrict the review to liposome mediated delivery of drugs and NAs, then the title should be altered accordingly.

We thank the reviewer for overall positive evaluation of our manuscript as well as for stylistic and grammar corrections. We altered the title and corrected the text accordingly to the file attached. 

Reviewer 2 Report

The manuscript written by Shmendel et al. is about summarizing the role of folic acid (FA) in the presence of lipid nanosystems for cancer therapy. The authors have shown different strategies: (i) a covalent approach in which FA has been introduced modifying different positions of a siRNA’s sense strand; (ii) an electrostatic approach; and (ii) FA lipoconjugates modifying liposome content.

The manuscript is interesting and contains a good number of recent references. I think the manuscript’s topic falls within the journal’s scope. However, I find several issues that the authors should address in order to consider this manuscript for publication in Pharmaceutics.

Major comments

(1) Page 2. Line 78. Conjugation of FA to nucleic acids. The authors should describe in depth how these conjugations were carried out. For example, is there involve a 1,3-cycloaddition, etc.? In addition, in Figure 2,  I recommend including the chemical entities involved in the conjugation between FA and the siRNA’s sense strands.

(2) The authors should include a new section summarizing the cellular internalization mechanisms of folate conjugates including different types of administration routes.

(3) Tables. It would be beneficial to add a new column to all tables that displays the pertinent results of the chosen reference.

(4) The authors should explain/discuss in depth the biological activities of the selected nanosystems both in vitro and in vivo, putting more emphasis on in vivo activities

(5) The authors should include the use of exosomes + FA in the manuscript

(6) Please, add a new section that describes the theragnostic applications of folate-targeted nanosystems in cancer therapy.

(7) The authors have introduced other polymers, like cyclodextrins or oligochitosan. I suggest including some results involving silica NPs and graphene.

(8) The manuscript should also contain a final section in which the authors describe future perspectives/personal overview of this type of strategy.

Author Response

Dear Reviewers,

Thank you very much for profound analysis of our manuscript. According to your comments we carefully revised the article.

Reviewer 2

(1) Page 2. Line 78. Conjugation of FA to nucleic acids. The authors should describe in depth how these conjugations were carried out. For example, is there involve a 1,3-cycloaddition, etc.? In addition, in Figure 2,  I recommend including the chemical entities involved in the conjugation between FA and the siRNA’s sense strands.

Following the advice we added information about conjugation methods in the text and added chemical structures in Figure 2.

(2) The authors should include a new section summarizing the cellular internalization mechanisms of folate conjugates including different types of administration routes.

Internalization mechanisms are widely discussed elsewhere (Cheung A. et al. Oncotarget 2016, 7, 52553; Rana A. et al. Bioorg. Chem. 2021, 112, 104946), but following the advice of the reviewer we added a brief description with references (page 2). Now it reads, “FA internalized by cells through the reduced folate carrier (RFC), the proton-coupled folate transporter (PCFT) or FR. Only the last one is capable of binding FA conjugates and transport them by receptor-mediated endocytosis”.

(3) Tables. It would be beneficial to add a new column to all tables that displays the pertinent results of the chosen reference.

Corrected as suggested (Tables 2 and 3).

(4) The authors should explain/discuss in depth the biological activities of the selected nanosystems both in vitro and in vivo, putting more emphasis on in vivo activities

We thank the reviewer for this suggestion. In vivo biological activities are indeed very important features of delivery systems, given in this manuscript. Additional discussion was added throughout the manuscript as suggested (in Section 4.2 and Section 4.3).

(5) The authors should include the use of exosomes + FA in the manuscript

We thank the reviewer for his criticism. Use of exosomes is rather new side of delivery systems. Following the advice of the reviewer we briefly discussed this issue in the Conclusions and Perspective section.

(6) Please, add a new section that describes the theragnostic applications of folate-targeted nanosystems in cancer therapy.

Theragnostic applications also included in the Conclusions and Perspective section (Page 13). However, we would like to emphasize that the present review focuses only on the design of folate-targeted liposomal systems for delivery of therapeutic nucleic acids.

(7) The authors have introduced other polymers, like cyclodextrins or oligochitosan. I suggest including some results involving silica NPs and graphene.

Several examples of cyclodextrins and oligochitosan derivatives were added to the amended version of the manuscript. Inorganic nanoparticles are far more different. We mentioned their application in the Introduction section (page 2), but we did not include information about polymeric and inorganic systems in the separate section. In 2021 these systemes were described in review "Folate receptor-mediated small interfering RNA delivery: recent developments and future directions for RNA interference therapeutics" (Nucleic acid research, 2021, doi: 10.1089/nat.2020.0882). Our manuscript summarises the latest data on liposomal delivery systems decorated with folate ligands.

(8) The manuscript should also contain a final section in which the authors describe future perspectives/personal overview of this type of strategy.

Corrected as suggested (Page 13).

Reviewer 3 Report

pharmaceutics-2283471

Nucleic acid delivery by folate-containing lipid nanosystems for antitumor therapy

The manuscript by Shmendel et al. summarized the role of folate in the delivery of nucleic acid using lipid nanosystems to treat cancer. The authors provided an interesting review of the topic. However, the manuscript can be improved following some comments below.

1. The Abstract should be rewritten to highlight the significance of folate-containing lipid nanosystems in nucleic acid delivery for antitumor therapy.

2. What is the coverage of this review? Lipid nanosystems or liposomes? It was lipid nanosystems in some places, whereas other parts mentioned liposomal systems. Please make them consistent.

3. Section 2 is superficial and should be expanded.

4. Tables 2 and 3: Please briefly include primary in vitro and in vivo outcomes.

5. Some parts should be deeply discussed to highlight the role of folate.

6. Please include a section to discuss the structure of lipid nanosystems, the localization of nucleic acid in the systems, as well as the effects of folate on these structures.

7. The number of related studies is relatively small. Can the authors find more studies of folate-containing lipid nanosystems?

Author Response

Dear Reviewers,

Thank you very much for profound analysis of our manuscript. According to your comments we carefully revised the article.

Reviewer 3

  1. The Abstract should be rewritten to highlight the significance of folate-containing lipid nanosystems in nucleic acid delivery for antitumor therapy.

We have modified the Abstract and added significance of folate-containing lipid nanosystems.

  1. What is the coverage of this review? Lipid nanosystems or liposomes? It was lipid nanosystems in some places, whereas other parts mentioned liposomal systems. Please make them consistent.

This review is focused on folate-containing liposomal delivery systems for nucleic acid delivery. Nevertheless, some other nanosystems were also included. We changed the title to avoid misunderstanding.

  1. Section 2 is superficial and should be expanded.

Corrected as suggested.

  1. Tables 2 and 3: Please briefly include primary in vitro and in vivo outcomes.

Column with results was added in Tables 2 and 3 as suggested.

  1. Some parts should be deeply discussed to highlight the role of folate.

Additional discussion was added throughout the manuscript as suggested (especially in Section 4.2 and Section 4.3)

  1. Please include a section to discuss the structure of lipid nanosystems, the localization of nucleic acid in the systems, as well as the effects of folate on these structures.

We do agree with the suggestion of the reviewer and thank him for this valuable advice. Localization of nucleic acid in nanoparticles is a wide and complex topic and should be addressed separately,  however, we have included in section 4.3 several paragraphs on the lipoplexes structures.

  1. The number of related studies is relatively small. Can the authors find more studies of folate-containing lipid nanosystems?

Literature pool was expanded as suggested. We focused on studies concerning folate-containing liposomal nanosystems for gene delivery published during the last 10 years. Other carriers are fully described in recent reviews https://doi.org/10.1089/nat.2020.0882  DOI: 10.1089/nat.2020.0893

Reviewer 4 Report

Maslove et al. have reviewer the general strategy for nuclei acid delivery by folate-containing lipi nanosystems for antitumor therapy based on the fact that cancer may overexpress FR on their surface and hence enhancing the FR-based targeting to delivery nucleic acids. The review is well written and informative. The following issues should be address before publications:

1.       Please add any clinical report for the regulated FR expression in patients with cancer

2.       Please include the discussion of advantages and disadvantages of each carrier

3.       Do table 2 reference contain any animal disease model?

4.       What is the fate of FA conjugate after transfecting it into cell?

5.       Please compare the colloidal stability of different carriers

6.       The introduction mentioned inorganic nanoparticles but the rest of the paragraphs did not discuss them

7.       Positive charges liposomes are more toxic. How to overcome it?

Minor:

1.       Line 109, 1 мМ? Should be  1 mM

2.       Some other inappropriate formats should be revised.

Author Response

Dear Reviewers,

Thank you very much for profound analysis of our manuscript. According to your comments we carefully revised the article.

Reviewer 4

  1. Please add any clinical report for the regulated FR expression in patients with cancer

We thank the reviewer for this valuable advice. As mentioned in the Introduction, tumour tissue from seventy-one percent of patients with triple negative breast cancer showed FR overexpression (Norton N. et al. npj Breast Cancer 2020, 6, 4). Some other examples of both in vitro and clinical studies are presented in the Introduction section.

  1. Please include the discussion of advantages and disadvantages of each carrier

Transfection efficiency and side effects of carrier depended on the target cell or tissue as well as on the chemical composition. We focused on liposomal carriers which composed of natural add synthetic lipids and lipoconjugates and posessed the similar side effect. Advantages/disadvantages of polymeric and inorganic systems were recentlu described in the review "Folate Receptor-Mediated siRNA Delivery: Recent Developments and Future Directions for RNAi Therapeutics" https://doi.org/10.1089/nat.2020.0882.

  1. Do table 2 reference contain any animal disease model?

Table 2 is focused on the optimization of the content of folic acid lipoconjugates. This optimization usually performed in vitro, so animal models are not included in this table, but were included in Table 3.

  1. What is the fate of FA conjugate after transfecting it into cell?

We added information about cellular internalization of folate-containing nanosystems (Page 2).

  1. Please compare the colloidal stability of different carriers

Unfortunately, colloidal stability of different liposomal carriers is not always discussed in the articles, so a adequate comparison of these delivery systems would be incorrect.

  1. The introduction mentioned inorganic nanoparticles but the rest of the paragraphs did not discuss them

Introduction contains basic information about all delivery systems available. The main section of this review focused on folate-containing liposomal delivery systems for nucleic acid delivery. We altered the title to avoid misunderstanding.

  1. Positive charges liposomes are more toxic. How to overcome it?

Positive charge may be “masked” by folic acid and/or PEG. This moment is already discussed in the manuscript (Page 10).

Minor:

  1. Line 109, 1 мМ? Should be  1 mM

Corrected as suggested.

  1. Some other inappropriate formats should be revised.

Checked.

Round 2

Reviewer 1 Report

The new title is now more appropriate for the content of the review. A few  minor corrections to the revision have been listed for your assistance. I have attempted to reword the final sentence (line 443) to convey the meaning intended by the authors, without using 'anyway', which I feel is inappropriate here. I recommend  that the authors attend to the minor corrections and that the manuscript be published.

Author Response

We thank the reviewer for evaluation of our manuscript as well as for stylistic and grammar corrections.  Changes are highlighted in the text. According to the suggestion, we deleted the word “anyway” in the final sentence. The revised manuscript has been approved by all authors.

Reviewer 2 Report

I acknowledge the effort carried out by the authors on this revised manuscript. I recommend the publication of this article in Pharmaceutics

Author Response

We thank the reviewer for evaluation of our manuscript as well as for stylistic and grammar corrections.

Reviewer 3 Report

The manuscript was appropriately revised and can be accepted. There is only a minor point to correct. Some numbers, such as 1.00E+07, should be presented properly according to the journal guideline.

Author Response

We thank the reviewer for evaluation of our manuscript as well as for stylistic and grammar corrections.  Changes are highlighted in the text. . According to the suggestion of the Reviewer, we changed the format of some numbers in Tables 2 and 3. The revised manuscript has been approved by all authors.

Reviewer 4 Report

The authors have addressed my comments

Author Response

We thank the reviewer for evaluation of our manuscript as well as for stylistic corrections.